# North Atlantic variability and its links to European climate over the last 3000 years

Paola Moffa-Sánchez[1] & Ian R. Hall[1]

The subpolar North Atlantic is a key location for the Earth's climate system. In the Labrador Sea, intense winter air–sea heat exchange drives the formation of deep waters and the surface circulation of warm waters around the subpolar gyre. This process therefore has the ability to modulate the oceanic northward heat transport. Recent studies reveal decadal variability in the formation of Labrador Sea Water. Yet, crucially, its longer-term history and links with European climate remain limited. Here we present new decadally resolved marine proxy reconstructions, which suggest weakened Labrador Sea Water formation and gyre strength with similar timing to the centennial cold periods recorded in terrestrial climate archives and historical records over the last 3000 years. These new data support that subpolar North Atlantic circulation changes, likely forced by increased southward flow of Arctic waters, contributed to modulating the climate of Europe with important societal impacts as revealed in European history.

---

[1] School of Earth and Ocean Sciences, Cardiff University, Cardiff CF10 3YE, UK. Correspondence and requests for materials should be addressed to P.M.-S. (email: moffasanchezp1@cardiff.ac.uk)

Ocean circulation has a key role in the Earth's climate, as it is responsible for the transport of heat but also its storage in the ocean's interior. Because of the large heat capacity of water and hence its thermal inertia, the ocean is potentially amongst the most predictable components of the Earth's climate system on time scales from decades to centuries. The subpolar North Atlantic, specifically, is a key region for understanding climate variability, as it is one of the world's main areas of deep water formation. Here, northward flowing warm and salty surface waters lose their heat to the atmosphere, become denser and eventually sink, as part of the Atlantic Meridional Overturning Circulation (AMOC), forming around half of the global deep water[1]. This process is not only important for the ventilation of the oceans abyss but its attendant northward heat transport[1] contributes to shaping the climate of northwest Europe.

Changes in the strength of the deepwater formation process have widely been proposed as the mechanism behind multi-decadal sea surface temperature variability in the North Atlantic[2,3], which has been shown to have important impacts on atmospheric patterns and the weather in Europe[4,5]. Recent work indicates that specifically the strength of deep water formation in the Labrador Sea is key component for driving variability in the strength of the AMOC, and hence for modulating recent and also decadal North Atlantic climate variability[6,7]. Observational studies have shown that interannual to decadal changes in the formation of deepwater in the Labrador Sea, namely Labrador Sea Water (LSW), are driven by changes in the upper ocean density gradients controlled by heat removal from winds and/or buoyancy forcing from freshwater input[8]. However, because of the lack of oceanographic measurements beyond the last 100 years, our understanding of the oceans' role, particularly centennial changes in the strength of LSW formation and associated subpolar gyre strength, in European climate over longer time scales remains fairly limited.

Several model studies have suggested that centennial-scale climate variability in the North Atlantic over the current interglacial was largely driven by changes in the formation of LSW responding, albeit non-linearly, to freshwater inputs from the Arctic Ocean into the Labrador Sea[9,10]. Centennial timescale increases in the export of polar waters into the subpolar North Atlantic spanning the last 10,000 years, have been recorded in the abundance of ice-rafted debris deposited in marine sediment cores[11]. These records have been widely used to establish a temporal framework for cold climatic events recorded in the circum-North Atlantic region by invoking ocean–land linkages[12]. Yet, there are very limited data that support the mechanism by which these pulses of ice-laden, fresh, Arctic Ocean waters impacted the ocean circulation in the North Atlantic and specifically, the strength of LSW formation and the surface circulation around the subpolar gyre, which are very likely candidates for the modulation of the northwest European regional climate. This is largely because of the lack of high sediment accumulation sites for proxy reconstructions at key locations, such as the areas of active deep water formation in the centre of the Labrador Sea.

In this study, we present a suite of subdecadally to decadally resolved proxy records from across the subpolar North Atlantic from which we can infer changes in the formation of deep waters in the Labrador Sea and its associated gyre strength across the last 3000 years. This interval spans several important periods within European history, which have often been related to climate variability such as the warm intervals during the Roman Empire expansion (colloquially referred to the Roman Warm Period ~250 years Before Common Era (BCE)—400 years Common Era (CE)) and Medieval times (Medieval Climatic Anomaly ~900–1200 years CE) and the cold periods such as the one centred around ~2700 years Before Present (BP) known as the

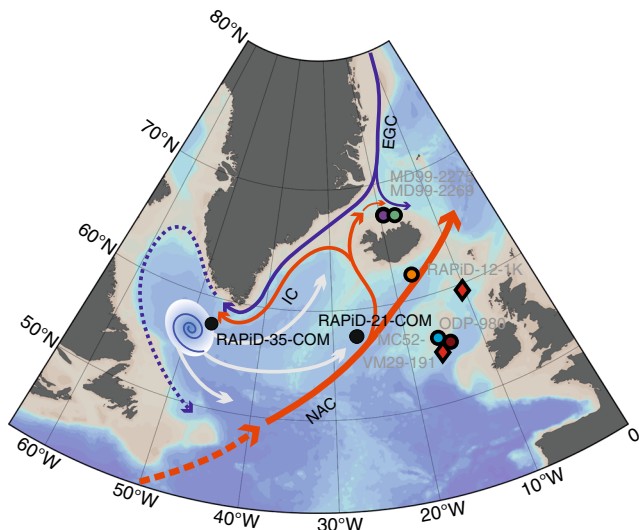

**Fig. 1** Sediment core location and regional ocean circulation. Red arrows indicate the warm and salty waters originating from the North Atlantic Current (NAC) flowing west as the Irminger Current (IC). Cold and fresh polar waters from the East Greenland Current (EGC) are indicated by the dark blue arrows, the dotted blue arrow indicates the West Greenland Current. The locus of Labrador Sea Water (LSW) formation is indicated by the blue spiral and the white arrows indicate the spreading of LSW through intermediate depths to the Irminger and Iceland Basins and to the lower latitudes. New reconstructions used in this study are shown in black and location of published proxy records presented in Figs. 2 and 3 are colour-coded and labelled in grey. Unlabelled red diamonds show the locations of the deep sea corals from ref. [42]. Bathymetric basemap made using ODV (Schlitzer, R., Ocean Data View, https://odv.awi.de, 2015)

Iron Age Cold Epoch, the short-lived Dark Ages Cold Period (~500–750 years CE) and the Little Ice Age (~1450–1850 years CE). Studying the ocean changes over the last 3000 years at high temporal resolution thereby provides a unique opportunity to investigate the potential linkages between ocean circulation changes and European climate variability and its impacts on societies. Our new findings suggest centennial changes in the circulation of the subpolar North Atlantic, likely modulated by the input of Arctic Ocean waters to the Labrador Sea, with similar timing to climate variability recorded on land in historical and terrestrial proxy data in Europe for the last 3000 years.

## Results

**Marine sediment cores.** Two high-resolution marine sediment cores located in the South Greenland Margin and South of Iceland were used to infer changes in the subpolar North Atlantic circulation over the last 3000 years. Composite record RAPiD-35-COM comprising box-core RAPiD-35–25B[13] (57°30.47′ N, 48° 43.40′ W, 3486 m water depth) and piston core RAPiD-35-14P (57° 30.250′ N, 48° 43.340′ W, 3484 m water depth) is located in the eastern Labrador Sea on the Eirik Drift (Fig. 1). This composite sediment record is in an ideal location to monitor shifts in the polar front which separates the warm and salty North Atlantic Current derived waters of the Irminger Current (IC) and the southward flowing fresh and cold polar waters of the East Greenland Current (EGC) (Fig. 1)[14] and hence the input of polar waters into the Labrador Sea. The core-chronology is presented in ref. [15], and indicates an average temporal sample resolution of 20–25 years. In addition, RAPiD-21-COM is a composite sedimentary record comprising box-core RAPiD-21-12B and kasten core RAPiD-21-3K (57° 27.09′ N, 27° 54.24′ W and 57° 27.09′ N, 27° 54.53′ W, respectively; at 2630 m water depth). RAPiD-21-

COM cores were recovered from the southern limb of the Gardar Drift (Fig. 1) at a location where the near-bottom flow speeds have been shown to be modified according to the volume of LSW reaching the Iceland Basin[16]. The chronology for RAPiD-21-COM is described in detail in ref. [17] and suggests an average temporal sample resolution of ~7.6 years.

**Changes in the input of polar waters into the Labrador Sea.** Model studies have shown correlation between LSW thickness and surface density[18]. Specifically, salinity exerts a dominant control on the upper-ocean density, which is driven by the relative input between the salty IC waters and freshwater and sea ice from EGC[19–21]. During the spring-summer months, after the winter convection has ceased in the Labrador Sea, its northeast boundary currents (the EGC and IC) support restratification of the surface ocean through lateral transport. The advection of heat and salt by these currents into the centre of the Labrador Sea has a critical role in the preconditioning of the water column for winter convection[22]. The RAPiD-35-COM site, is therefore ideally located to study past alterations of the surface buoyancy forcing in the Labrador Sea as a hydrographic preconditioning for deep water formation by monitoring changes in the relative presence of these two different waters (EGC and IC) reaching the eastern Labrador Sea[13]. To reconstruct the relative influence between the fresh and cold polar waters from the EGC and the warm and salty waters of the IC in the eastern Labrador Sea during spring-summer restratification we use two different proxies comprising planktonic foraminiferal assemblages and $\delta^{18}O$ composition.

We measured the oxygen isotopes from planktonic foraminifera which are marine unicellular calcifying organisms that live in the surface ocean waters. Specifically, we measured two different species from RAPiD-35-COM, the polar *Neogloboquadrina pachyderma* (sinistral coiling) (Nps) and the subpolar *Turborotalita quinqueloba* (Tq). A core-top study from a longitudinal transect across the Nordic Seas[23] found that the calcification depth of these two species differs according to the hydrographic conditions at the site. A constant near-surface calcification depth was found for Tq (25–75 m) across this region[23]. In contrast, in the eastern section of the Nordic Seas, under the presence of warm Atlantic waters of the Norwegian Current, Nps was found to calcify deeper in the water column (100–200 m), whereas in the west under the influence of the EGC polar waters it calcified closer to the surface at a similar depth as Tq[23]. Following these findings, we used the difference in the $\delta^{18}O$ composition between Nps and Tq (Supplementary Fig. 1), referred hereafter as $\Delta\delta^{18}O_{Nps-Tq}$, as an indicator of the relative presence of warm Atlantic waters influencing the RAPiD-35-COM site in the past (See 'Methods' section). Large/small differences in $\Delta\delta^{18}O_{Nps-Tq}$ indicating increased/decreased presence of warm and salty Atlantic IC waters vs. polar EGC waters in the upper water column, respectively. In addition, an independent measure of the relative presence of polar waters in the eastern Labrador Sea can also be gained by using the percentage abundance of the polar species Nps. The distribution of Nps abundances in modern sediments show the affinity of this species to cold polar surface waters[24] and it also exhibits large abundance changes across oceanic fronts[25]. For this reason, Nps abundance is a widely used proxy to track variability in the position of the polar front across different time scales[26]. However, both of these proxies are largely dominated by temperature whereas it is the buoyancy forcing and hence largely the haline component that is the most important for the modification of LSW formation[19–21]. Thus, we use a 70-year long observational time-series from the Labrador Sea region to show the strong positive correlation ($R^2 = 0.86$, Supplementary

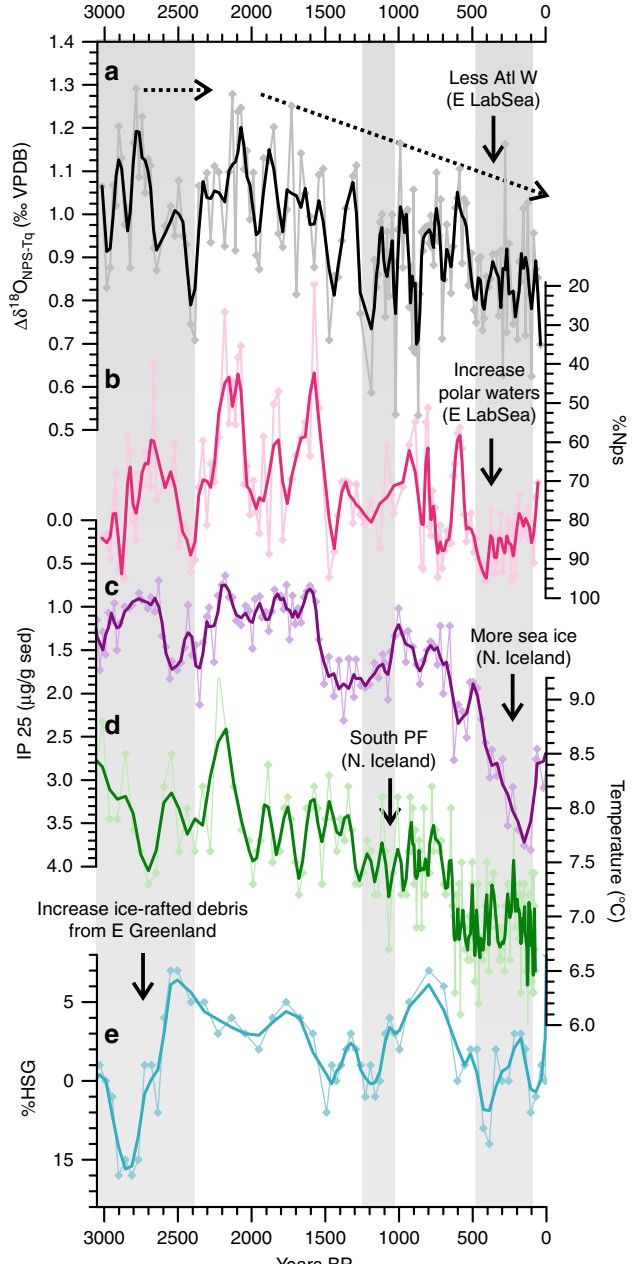

**Fig. 2** Changes in the southward influence of polar waters. **a** Difference in the $\delta^{18}O$ measurements of *N. pachyderma* (s) and *T. quinqueloba* from RAPiD-35-COM as an indicator of the relative presence of Atlantic waters in the Eastern Labrador Sea (data between 0 and 1250 years BP presented in ref. [13]). **b** Percentage of the polar species *N. pachyderma* (s) from RAPiD-35-COM (data between 0 and 1250 years BP presented in ref. [13]). **c** Sea ice reconstructions based on IP25 proxy from core MD99-2269[28] located North of Iceland (Fig. 1). **d** Diatom assemblage derived August temperatures from North of Iceland[27]. **e** Percentage of haematite stained quartz grains (HSG) from MC52-VM29-191[11]. Bold lines indicate weighted three-point running average and grey bars indicate the periods of glacier advances in Alaska and Swedish Lapland[49]

Note 1; Supplementary Figs. 2 and 3) between temperature and salinity in the top 150 m at decadal time scales. This relationship adds confidence to our proxy interpretation that reconstructed cold ocean conditions were likely also fresh.

Results from RAPiD-35-COM within the eastern Labrador Sea show coherent changes in the surface ocean conditions in the two

complementary proxies ($\Delta\delta^{18}O_{Nps-Tq}$ and %Nps) (Fig. 2a, b). The $\Delta\delta^{18}O_{Nps-Tq}$ shows smaller values, hence reduced influence of Atlantic waters mostly coincident with intervals of increased abundance of Nps, suggesting an increased influence of polar waters and hence a southern advancement of the polar front position at around 2700–2300 years BP, 1500–1000 years BP and 500–100 years BP (Fig. 2a, b). Although there are other plausible oceanic processes which could explain the cooling recorded in the proxies at the RAPiD-35-COM site such as increase in deep convection and cooling of the surface waters, the magnitude of variability recorded by the %Nps (approximately equivalent to 2°C[24]), would be difficult to account for without invoking an influence of frontal shifts.

Paleoceanographic reconstructions from a more northward location of the polar front on the North Iceland margin (Fig. 1), show centennial-scale cold events[27] (Fig. 2d) and marked increases in sea ice[28] (Fig. 2c) with similar timing to the cold events recorded in the eastern Labrador Sea (Fig. 2a, b). Furthermore, these cold events coincide with the increase in the abundance of haematite stained quartz grains found in core MC52-VM29-191 located in the Rockall Trough[11] (Fig. 1, 2e). The provenance of these ice-rafted grains has been shown to be specific to northeast Greenland and hence consistent with an increase in the southern transport of drift ice within the EGC and

around the subpolar gyre to the core site[11]. These data collectively indicate an increase in the influence of ice-laden fresh and cold EGC waters and a southern migration of the polar front in the subpolar North Atlantic during 2700–2300 years BP, 1500–1000 years BP and 550–100 years BP (Fig. 2). Conversely, increased $\Delta\delta^{18}O_{Nps-Tq}$ and reduced %Nps between 2300–1500 years BP and 900–550 years BP consistently suggest periods of enhanced influence of warm (and salty) Atlantic waters in the Eastern Labrador Sea at these times (Fig. 2a, b). Additional paleoceanographic records from around Greenland (including for example Fram Strait[29], East Greenland[30], Denmark Strait[31], and West Greenland[32,33], and references therein), consistently show a millennial increase in the influence of polar EGC waters and drift and sea ice over the last ~2500 years coherent with the trends shown in Fig. 2. Albeit the lower temporal resolution and the associated dating uncertainties, some of these records record centennial ocean fluctuations, with an increased influence of Atlantic waters around Greenland between ~1400–2400 years BP[30–32] and an increase influence of cold polar ice-laden EGC waters at ~2500–2700 years BP[31–33] and during the LIA[31,32,34,35].

**Labrador Sea Water and Overflow Water interactions.** Surface freshening has a primary role in inhibiting convection in the Labrador Sea, as identified during, for example, the 1970s Great Salinity Anomaly[21]. Similarly, our recorded centennial increase in cold and fresh polar waters reaching the eastern Labrador Sea would have led to stratification of the upper water column, likely also limiting winter deep water formation in the Labrador Sea. However, because of the intermediate nature of LSW depth[8] and the absence of suitable coring sites directly bathed by this water mass, confirmation of past changes in the formation of LSW using marine sediment cores is challenging. To address this, we compare the data from the eastern Labrador Sea with a new record from RAPiD-21-COM, that albeit indirectly, allow us to infer changes in LSW formation through its interaction with the

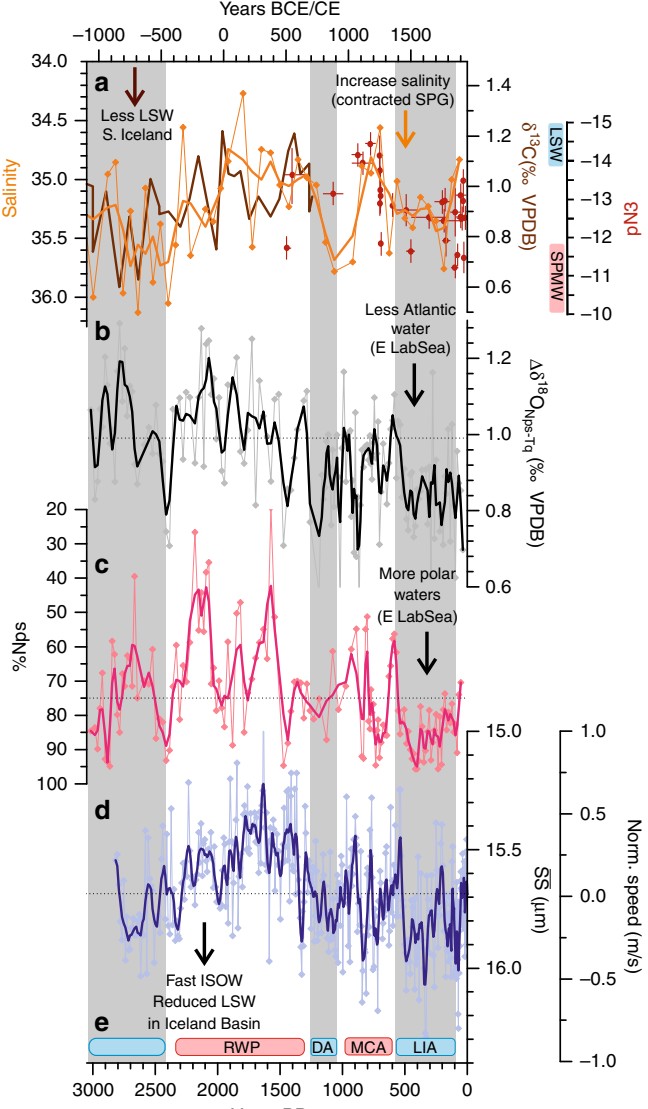

**Fig. 3** Indirect indicators for Labrador Sea Water changes. **a** $\delta^{13}C$ data from ODP Site-980[40] indicated by the brown line, the depth of this core is 2170 m and hence lies on the lower boundary of LSW, waters with a $\delta^{13}C$ signature closer to modern LSW (~1.3 ‰) are found during warmer climatic periods. The compilation of $\epsilon$Nd data from deep sea corals recovered between 630 and 1325 m[42] shown as red data points. The vertical red and blue bars on the right-hand axis indicate the approximate $\epsilon$Nd-signature for LSW and Subpolar Mode Waters (SPMW). Superimposed in orange is the salinity data from South of Iceland from RAPiD-12-1K[47] interpreted to be an indicator of contracted/weaker and expanded/stronger gyre. **b** Difference in the $\delta^{18}O$ measurements of *N. pachyderma* (s) and *T. quinqueloba* from RAPiD-35-COM as an indicator of the relative presence of Atlantic waters in the eastern Labrador Sea (data between 0 and 1250 years BP presented in ref. [13]). **c** Percentage of the polar species *N. pachyderma (s)* from RAPiD-35-COM (data between 0 and 1250 years BP presented in ref. [13]). **d** $\overline{SS}$ from RAPiD-21-COM, faster speeds of the deep current from the Iceland-Scotland Overflow indicative of a decreased LSW volume in the Iceland Basin, normalised near-bottom current speeds were calculated using the calibration from ref. [39](data between 0 and 166 years BP presented in ref. [16]). Bold lines indicate the weighted three-point smoothed data and horizontal dotted lines show the average of the records for the last 3000 years, **e** Red and blue rectangles indicate the cold and warm periods recorded from glacier advances and retreats in Alaska and Swedish Lapland[49] (for more discussion on glacier dynamics refer to Supplementary Fig. 4) and grey bars highlight only the glacier advances. Acronyms refer to the historical periods with similar timings to the climatic periods, Roman Warm Period (RWP), Dark Ages Cold Period (DACP), Medieval Climatic Anomaly (MCA) and Little Ice Age (LIA)

strength of the Iceland-Scotland Overflow Water (ISOW). Although RAPiD-21-COM lies in the pathway of the ISOW, it is ~900 km downstream of the Iceland-Scotland Ridge and therefore, the overflow waters that reach the site will have undergone significant modification through entrainment and mixing with the LSW[36]. In the Iceland Basin, LSW is present at mid-depths with greater thickness during periods of increased deep water formation in the Labrador Sea[37]. A subdecadally resolved near-bottom flow speed reconstruction spanning the last 230 years from the box-core recovered at the same location as RAPiD-21-COM, suggested that ISOW strength at this site can be strongly modulated by the volume/density of the overlying LSW in the Iceland Basin[16]. Thus, for example, periods of reduced LSW presence at intermediate depths in the Iceland Basin, such as in the 1960's, corresponded to faster ISOW flow speeds, vs. slower ISOW during periods of enhanced LSW presence during 1994[16].

In light of these results, we extended the 230-year near-bottom flow speed record[16] using RAPiD-21-COM, for the last 3000 years (Fig. 1) to gain an indirect and qualitative understanding of the influence of LSW in the Iceland Basin, through its interaction with the strength of ISOW flow speeds. We used the near-bottom flow speed proxy sortable silt mean grain size ($\overline{SS}$), the mean grain size of the 10–63 μm terrigenous material[38] (see 'Methods' section). The new $\overline{SS}$ data from site RAPiD-21-COM show very clear variability which according to a recent calibration[39] equate to a maximum variance in the ISOW flow speed of ~2 cm/s (Fig. 3d). Generally faster near-bottom flow speeds (grain sizes above the average for the 3000-year interval of 15.7 μm, Fig. 3) are found during three intervals around 3000–2500, 1200–1000 and 550–100 years BP (Fig. 3d). Following ref. [16], strong ISOW flow speeds would indicate a reduced presence of LSW in the Iceland Basin, which has been associated with weak deep water formation in the Labrador Sea[8]. The new $\overline{SS}$ results from RAPiD-21-COM show clear centennial variability with periods of faster speeds broadly corresponding to the centennial cold conditions recorded in the eastern Labrador Sea (Fig. 3b, c). The Iceland Basin results are therefore consistent with the suggestion that the advancement of the polar front and an increase in the influence of cold/fresh Arctic water input into the eastern Labrador Sea likely reduced the formation of LSW at these times. In contrast, the millennial timescale variability appears to differ between the proxy records. The records from the northernmost sites (Fig. 2c, d), show a linear cooling trend perhaps driven by the Neoglacial decrease in summer insolation in the northern high latitudes and its effects on Arctic sea ice production. This long-term cooling trend is also present in the $\Delta\delta^{18}O_{Nps-Tq}$ (Fig. 2a) and differs from the Nps perhaps due to slight change in the timing of the seasonal bloom of the two foraminiferal species as a response to changes in insolation (Supplementary Fig. 1).

**Intermediate water mass signatures South of Iceland**. As additional support for the suggested changes in LSW formation we also use published benthic foraminifera $\delta^{13}C$ record from Ocean Drilling Program (ODP) Site-980 located South of Iceland[40] (Fig. 1). The $\delta^{13}C$ of the dissolved inorganic carbon in the ocean has a very specific distribution and can be used as a water mass signature. A recent study that accounts for the addition of anthropogenically derived $CO_2$ which contains isotopically light carbon, also known as the Suess effect, shows very distinct $\delta^{13}C$ distributions in the subpolar North Atlantic with a LSW $\delta^{13}C$ signature of 1.2–1.4 ‰; and a sharp boundary with the underlying North East Atlantic Deep Water ($\delta^{13}C$: 0.9–1 ‰)[41]. In light of these findings, ODP Site-980 located at 2179 m depth is therefore at a sensitive location to monitor past changes in the thickness/depth and hence ventilation associated with LSW[41], with lower $\delta^{13}C$ values suggesting a shallower/thinner and hence

weak LSW formation. Unfortunately, the ODP Site-980 $\delta^{13}C$ record (Fig. 3a) does not span the entire last 3000 years, but it does show broadly a transition from lower $\delta^{13}C$ around 3000–2500 years BP to higher values around 2000–1500 years BP (Fig. 3a), consistent with our new proxy record suggesting this interval was characterised by a shift from weaker to stronger LSW convection as inferred from the reduced ISOW flow speeds recorded in the $\overline{SS}$ from RAPiD-21-COM. As the ODP Site-980 $\delta^{13}C$ record only reaches 1300 years BP, we also compare published εNd data measured in deep sea corals from two nearby sites recovered from water depths between 635 and 1300 m[42] (Fig. 1). In this study, ref. [42] shows a shift in the εNd water mass signature from a predominantly LSW to a modified Atlantic water mass signature around 400 years BP (Figure 3a) again consistent with the transition we observe in the surface hydrography of the eastern Labrador Sea (Fig. 3b,c) and ISOW strength (Figure 3d) at the onset of the Little Ice Age at ~500 years BP.

**Changes in the subpolar gyre dynamics**. The process of convection associated with LSW formation is driven by intense surface heat lost during the winter months. This leads to a doming of isopycnals in the central Labrador Sea and results in an increase in the zonal density gradient across the subpolar gyre driving baroclinic circulation[43,44]. It also potentially modifies the shape of the subpolar gyre and the location of its associated fronts[45,46]. We exploit this connection to test whether periods of increased in polar water influence in the eastern Labrador Sea indeed weakened the deepwater formation in the region and also the strength of the gyre circulation. For this, we compare our results to published subpolar frontal shifts southeast of Iceland inferred from salinity reconstructions in core RAPiD-12-1K[47] (Fig. 1). Following the work by ref. [45], saltier surface conditions south of Iceland are suggested to reflect a weak and therefore contracted gyre and north-westward migration of the subpolar front. Conversely, fresher conditions south of Iceland are suggested to correspond to an expanded/stronger gyre[45]. Despite the lower temporal resolution of the South of Iceland salinity record[47], these data consistently show saltier (and warmer) conditions indicative of a contracted and weaker gyre (Fig. 3a) during periods of cold conditions in the eastern Labrador Sea (Fig. 3b, c) and reduced LSW formation (Fig. 3a, d).

**Land–ocean–atmospheric linkages**. Comparison of our new and published records not only reveal consistent timing of ocean variability in the subpolar North Atlantic across different sites, but also show a clear temporal correspondence with continental climate reconstructions and historical records (Figs. 3e, 4). Periods of increased influence of polar waters in the eastern Labrador Sea (Fig. 3b, c), reduced LSW formation (resulting in faster ISOW, Fig. 3d) and weaker subpolar gyre (Fig. 3a) largely coincide with well-established cold periods recorded in glacier advances, tree-ring and pollen records in the circum-North Atlantic and northwest Europe[48–51] (Fig. 3e; Supplementary Fig. 4). These cold periods have been associated with historical events of societal relevance such as famines and pandemics due to crop failures during the relatively short-lived Cold Ages Dark Period (recently renamed as the Late Ages Little Ice Age[52]) and the demise of the Norse settlements in Greenland at the onset of the Little Ice Age (LIA) with the consecutive complete isolation of Greenland by surrounding sea ice for the following centuries (Fig. 4). Conversely, periods of reduced influence of polar waters in the eastern Labrador Sea (Fig. 3b, c), stronger subpolar gyre (Fig. 3a) and increase LSW formation (resulting in slower ISOW, Fig. 3d) largely coincide with mild/warm periods in Europe namely the Roman Warm Period and the Medieval Climatic Anomaly. These intervals are recorded as periods of glacier

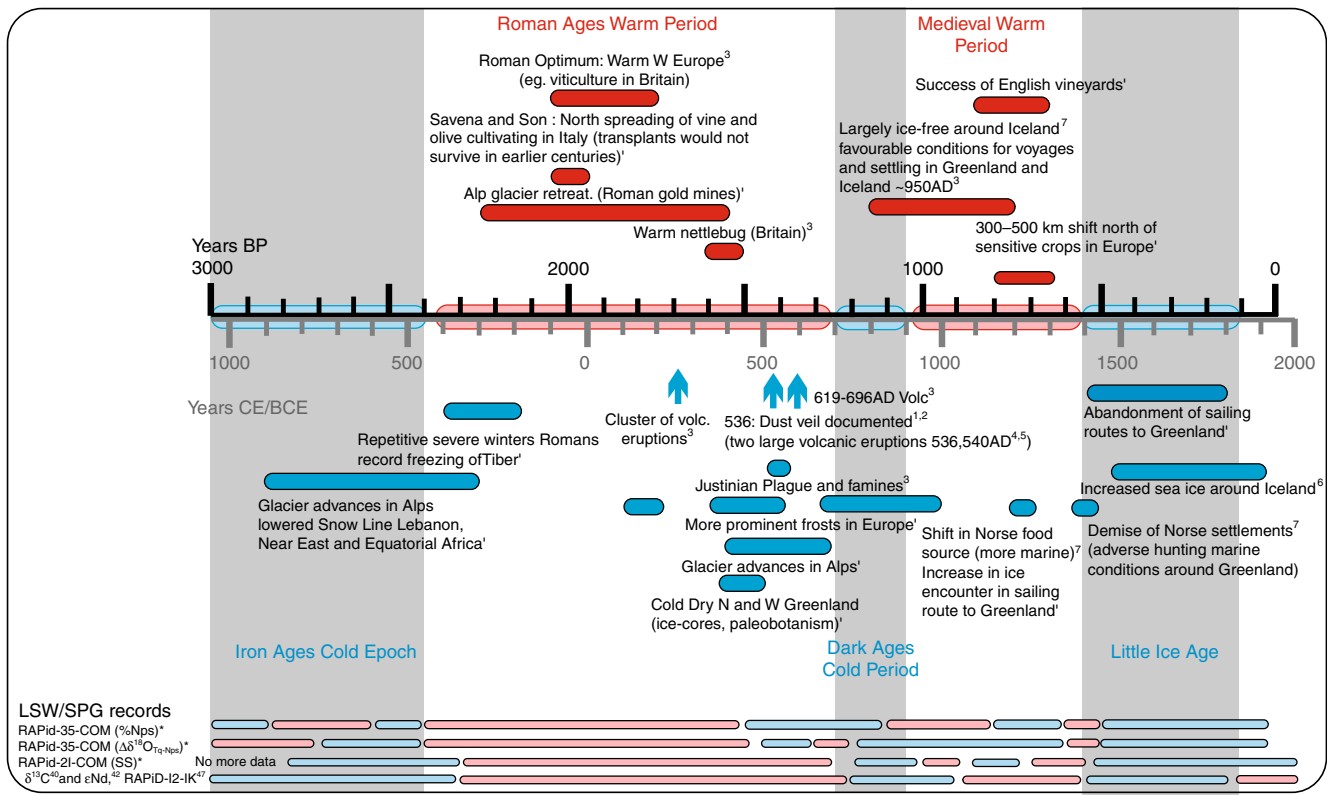

**Fig. 4** Schematic timeline highlighting historic records of climate variability in Europe. Red and blue lines denote the time-span for the evidence for warm and cold periods, respectively. Ages are in years BP (black) and years CE/BCE (grey). This information has been extracted from several publications indicated by the superscript in the annotations: 1. ref. [68] and references herein; 2. ref. [69]; 3. ref. [70] and references herein; 4. ref. [71]; 5. ref. [72]; 6. ref. [73]; 7. ref. [74] The cold and warm periods established through the glacier advances and retreats used as a framework for the study of these centennial events[49] are found within the axis of the timeline and highlighted by the vertical grey bars (consistent with Figs. 2 and 3). The marine paleoceanographic reconstructions for the LSW and Subpolar Gyre (SPG) presented in Fig. 3 are represented in blue and pink horizontal bars indicating time intervals below and above average values of the records for the last 3000 years for weaker and stronger LSW/SPG, respectively. For more information on the agreement with terrestrial proxy records and historical events see Supplementary Fig. 4

retreats (Fig. 3e)[49] and/or periods of no glacier advances (Supplementary Fig. 4), with milder temperatures allowing the northward expansion of vineyard crops to the North of Italy and even the British Isles (Fig. 4). This strong correspondence in the timing of ocean and continental climate variability suggests that the ocean conditions, particularly the formation of LSW, related changes in the subpolar gyre strength, and attendant northward heat transport, were probably key in modulating the climate in northwest Europe (Fig. 5).

The overriding question is: what drove these shifts in the subpolar North Atlantic? We present convincing evidence for the importance of the southward export of polar, sea ice laden waters, into the North Atlantic delivering the freshwater from the Arctic Ocean into the Labrador Sea. However, variability in the heat removal from the central Labrador Sea via wind stress changes could have also had a role in driving the recorded ocean variability. For instance, a weaker and more meridional westerly winds (associated with a persistent negative North Atlantic Oscillation, NAO) would have not only limited heat loss from the surface of the central Labrador Sea reducing deep water formation and weakening the gyre strength, but would have also enhanced the southward transport of polar waters from the EGC and vice versa under persistent positive NAO-like conditions.

Atmospheric circulation reconstructions across the last 3000 years are limited and show differing results. Yet, several studies suggest periods of a predominant negative NAO during 3000–2500 years BP and the LIA (~500 years BP) with generally more positive NAO-like circulation around 500–2000 years BP[53–]

[56], which would broadly correspond with the timing in the ocean variability presented in Fig. 3. However, most of these records use single or bi-proxy environmental records for NAO reconstructions, a methodology that has been recently questioned to yield robust results[57]. For example, if we focus on the last millennium, where there is more data available, a new statistical method applied to a network of 48 annually resolved proxy records has found no evidence for a persistent negative NAO conditions during the LIA[57], supporting modelling studies that suggest no significant atmospheric change at this time[58]. Similarly, to the LIA, it may be that the NAO did not undergo long-lasting centennial shifts during the climatic events of the last 3000 years. This would be in agreement with atmospheric variability exerting a more dominant control on the ocean over shorter time scales (decadal)[59], whereas the longer centennial timescale changes likely being forced by the ocean through small buoyancy changes in central Labrador Sea[58], also perhaps impacting the atmospheric circulation as a result. If this is the case, an increase in the southward transport of polar waters into the Labrador Sea alone would have been sufficient to drive the reconstructed ocean variability in the LSW formation and gyre strength, as suggested by last millennium model studies[60,61].

Our data provide evidence of concomitant centennial-scale changes in the subpolar North Atlantic circulation and northwest European climate over the last 3000 years. Yet, whether these changes were associated with an overall AMOC reduction during these centennial events still remains equivocal. The Nordic Seas Overflows contribute with the densest waters to the deep limb of

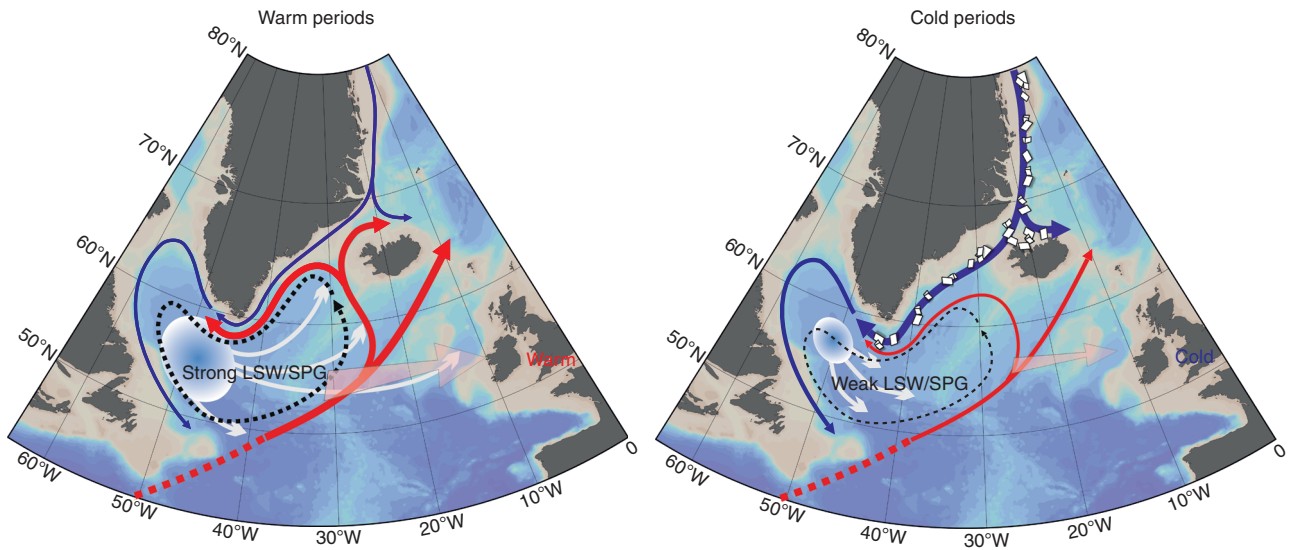

**Fig. 5** Schematic of the ocean circulation patterns during centennial warm and cold periods. The Atlantic waters (from the North Atlantic Current and the IC) and Polar Waters (EGC) are represented in red and blue arrows, respectively with the thickness of the lines representing the contribution at the northern boundary of the Subpolar Gyre (SPG) and transport. The black dotted lines indicate the SPG and the white circle and the arrows indicate the formation and spreading of LSW. The faded pink arrow represents the heat transport towards Europe in each scenario. Bathymetric basemap made using ODV (Schlitzer, R., Ocean Data View, https://odv.awi.de, 2015)

the AMOC[62], but these do not show coherent centennial changes in their flow speed over the last 3000 years[15], and, in fact, their multicentennial scale variability suggests compensating transport between the overflows east and west of Iceland (Faroe Bank Channel and Denmark Strait, respectively)[15]. Although it is therefore probable that LSW formation had an important role in modulating centennial-scale changes in the AMOC strength, it is also entirely conceivable that these oceanic processes did not significantly reduce the AMOC and perhaps the climate of northwest Europe during the last 3000 years was mainly a response to changes in the oceanic northward heat transport predominantly controlled by subpolar gyre strength and the heat transport by its boundary currents. This mechanism is consistent with processes found in numerical model studies, which explain the onset of the LIA and its associated cold European winter temperatures as a result of changes in the subpolar gyre strength[61,63].

Observations and model studies coherently suggest a twentieth century freshening of the North Atlantic particularly in the Labrador Sea due to an accelerated melting of the Greenland Ice Sheet and Arctic run-off[64]. This freshening is already having effects on deep water formation in the Labrador Sea and the AMOC[65,66]. However, unlike the natural late Holocene oceanic variability presented in this study, the twentieth century freshening of the surface waters in the Labrador Sea is also accompanied by general warming, particularly of the IC waters, which will lead to a greater reduction in surface densities potentially further limiting convection in the Labrador Sea[18]. In addition, a recent model comparison study has highlighted the inability of most climate models to correctly represent the surface mixed layer depth in the subpolar gyre region and hence likely underestimating the potential for a future collapse of the LSW formation/subpolar gyre under enhanced freshwater forcing[67]. It is therefore essential that we continue to improve our understanding of the LSW/subpolar gyre dynamics at a range of time scales to reduce uncertainty in future climate predictions.

## Methods

**Planktonic foraminifera oxygen isotope measurements**. We measured the $\delta^{18}O$ of the foraminiferal calcite of approximately 40–70 *Turborotalita quinqueloba* (Tq) individuals and 50 *Neogloboquadrina pachyderma* (s) (Nps) individuals in the 150–212 μm size fraction in RAPiD-35-COM (Supplementary Fig. 1). Stable isotope measurements on the foraminiferal shells were performed on the Thermo Finnigan MAT 253 mass spectrometer coupled to a Kiel II carbonate preparation device at Cardiff University. The mass spectrometer was calibrated through the international standard NBS-19, and all isotopic results are reported as a per mil deviation from the Vienna Pee Dee Belemnite scale (‰VPDB). External reproducibility of carbonate standards was ±0.08 ‰ for $\delta^{18}O$.

**N. pachyderma (s) counts**. We estimated the %Nps in RAPiD-35-COM by splitting each sample and counting a minimum of 350 planktonic foraminiferal individuals between the 150–250 μm size fractions.

**Sortable silt mean grain size**. Sortable silt mean grain size ($\overline{SS}$) is the mean grain size of the 10–63 μm terrigenous material[38]. The size sorting of this particle size range behaves non-cohesively and thereby responds to hydrodynamic processes making it useful as a proxy for near-bottom flow speeds of its depositing current[38]. The preparation of the sediment from RAPiD-21-COM for $\overline{SS}$ analysis involved the removal of carbonate and biogenic opal using 2 M acetic acid and 0.2% sodium carbonate ($Na_2CO_3$) at 85 °C for 5 h. To ensure full disaggregation the samples were suspended in 0.2% Calgon (sodium hexametaphosphate) in 60 ml Nalgene bottles, and placed on a rotating wheel for a minimum of 24 h. Before measuring in a Beckman Multisizer 2 Coulter Counter the samples were finally ultrasonicated for 3 min. Each sample was measured three times with an average standard deviation between the average particle sizes of the three runs from the same sample of 0.19 μm. The splicing between RAPiD-21-12B and RAPiD-21-3K, into RAPiD-21-COM was done as described in ref. 17 using the % coarse fraction. The $\overline{SS}$ variability between the overlapping sections (35 cm) of the two cores (RAPiD-12-21B and RAPiD-21-3K) was in close agreement, although the $\overline{SS}$ values were consistently offset by 0.36 μm. To address this, the spliced $\overline{SS}$ record was constructed by applying an offset of +0.36 μm to RAPiD-21-3K.

**Data availability**. The data sets generated during the current study are available through the NOAA climate data centre (https://www.ncdc.noaa.gov/paleo/study/22790) and available from the corresponding author.

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

## Acknowledgements

We are grateful to the UK NERC and its Rapid Climate Change Programme for its support. We would like to thank Prof. Nick McCave and the crew of *RV Charles Darwin* 159 for the recovery of such unique sediment cores and Julia Becker and Karin Boessenkool for laboratory assistance. We thank Benjamin Jervis for informative discussions on the historical context, and Eduardo Moreno-Chamarro, Lukas Jonkers and Loïc Houpert for useful discussions on the manuscript. We are grateful to Igor Yashayaev for providing the hydrographic time-series from the Labrador Sea. We would like to also thank the three anonymous reviewers for their constructive comments that helped improve the manuscript.

## Author contributions

P.M.-S. performed the laboratory work and collected the data from RAPiD-35-COM. I.R.H. collected the data from RAPiD-21-COM. P.M.-S. led the compilation of the data, interpretation and writing of the manuscript. All authors contributed towards the data interpretation and generation of the manuscript.

## Additional information

**Competing interests:** The authors declare no competing financial interests.

