## [Peer Review File · Nature Communications]

Reviewers' comments:

Reviewer #1 (Remarks to the Author):

This study presents new proxy data from two marine sediments cores, which are used to reconstruct variability in the North Atlantic Subpolar Gyre during the late Holocene. The authors link influx of Arctic waters and Labrador Sea Water formation to climatic conditions in Europe. The results are compared to other relevant published records from the region, providing convincing evidence for the conclusions presented in this manuscript. The study of climate variability of the last millennia and its link to ocean dynamics is crucial for understanding present and future changes and this study makes a significant contribution. I can therefore recommend the publication of this article and have only very minor suggestions for revisions.

I suggest visualizing the two main scenarios that are discussed in the paper by including a conceptual figure/map. European cold period with polar waters in the E Lab Sea and reduced LSW formation vs. the warm periods with contrasting conditions. Each scenario can be shown on a map with currents and front positions as well as a profile across the main sites showing the water masses.

Line 42 Should refer to Figure 2e, not Figure 2d

Line 288 "Unlabeled red circles" to indicate corals locations are not shown on the map?

Line 320 Figure 4: include the marine records as in Figure S2.

Reviewer #2 (Remarks to the Author):

Moffa-Sanchez and Hall present an interesting study on decadal-scale hydrographic changes in the subpolar gyre during the period of last 3000 years BP and formation changes of deep water production in the Labrador Sea compared with Iceland-Scotland Overflow Water (ISOW) using planktonic foraminifer stable isotopes and sortable silt data.

The study is certainly of interest to improve the very little knowledge of the formation of Labrador Sea deep water and the strength of the northwest boundary of the subpolar gyre, yet the results could hardly reconstruct the behavior of the gyre at its center and consequences in its northeast boundary (along the path of the North Atlantic Drift).

Although, the study finds some links between major millennial-scale cold events within the last 3 ka and major historical events, the discussion at a decadal-scale of those links is somehow unsatisfactory.

The organization of the manuscript, presentation of the results and the general discussion need some improvements and revisions.

Specific comments on the manuscript:

I find the introductory paragraph very general, lacking specific characteristics of the study region. However, the reader could find explanatory sentences of specific oceanic setting along the manuscript, right before some results or discussions. The authors should consider moving those sentences to the intro paragraph.

As an example, lines 124-129, lines 145-153.

The records show far more variations (at a decadal scale) than the ones the authors refer to by the grey thick lines (figures 2 and 3): The decrease of Nps to 60-40% is brief, the following intervals

of increase are quicker even within the intervals that are identified here as being intervals of enhanced influence of warm water. how could the authors explain this? are there more variations than the ones the authors identify? are there variations within the main variations (more frequent frontal migrations)?

Looking at the location of the Site RAPid-12-1K, referred to as south of Iceland is a bit tricky and somehow misleading. The Site is located in the main path of the North Atlantic Drift, southeast of Iceland. The "behavior" of the subpolar gyre from the "perspective" of the site location is true (line 131-134), yet the comparison with the Study site raises some concerns. Even though both set of results show a clear similar trend, the S-E Labrador Sea results clearly tell another story embedded within the one that the S-E Iceland results are telling. The movements of the Polar-Arctic fronts are clearly more frequent south of the Labrador Sea than S-E Iceland where warm and salty water flows are much stronger. Is there an influence on the recovery/shut down of LSW formation?

Although the results of the second study site (21-3K) improve the discussion about deep water formation (ISOW) and the strength of the subpolar gyre (Salinity variations), they come in the manuscript far too late, I wonder if it's not best for the manuscript to introduce both sites at the same time, discuss the results and compare with other results at the same time, that would be more concise and clearer.

The Sortable Silt results are, in my opinion, not given enough attention as they represent a crucial factor in the discussion of ISOW flow speed and contribution of LSW, two or three sentences would be helpful to fully cover the results acquired. Furthermore, how did the authors deal with the influence of Ice rafted debris (direct sedimentation not representative of bottom sorting)? knowing that the study Site is close by Bond's MC52 site where influence of IRD has been recorded. I suggest having a look at the work of Hass, H. C. (2002), A method to reduce the influence of ice-rafted debris on a grain size record from northern Fram Strait, Arctic Ocean. Polar Research, 21: 299-306. doi:10.1111/j.1751-8369.2002.tb00084.x

The conclusion is drawn from the results and the discussion includes comparison with previous records. However, I'm not keen about the relevance of the general discussion involving the historical evolution of the European societies and the climatic influence on them. Two pages have been dedicated to the later discussion, setting apart the discussion of the climatic context (or mentioned rarely) that have been reconstructed and discussed above. I personally like those kinds of cross-examinations of Human evolution built on knowledge of environmental changes. Should the authors look for a more included discussion while discussing the study's results rather than leaving it to the very end? the answer is probably a matter of opinion. I suggest that figure 4 display the events (red and blue) along with the variations of Nps and d18O rather than with the three grey thick lines.

I am listing below my comments in order of the manuscript and not scientific weight.

line 42: correct Figure 2e instead of Figure 2d, referring to Ref 14

line 44: ...there are very limited data...

line 44: "confirm" is a strong verb; no need to add fresh before ice!; the sub polar gyre needs to be introduced beforehand

line 44-47: the whole sentence needs to be rewritten and broken down in many sentences if possible.

line 47-49: redundancy: the lack of data has already been mentioned at the beginning of line 44

line 50: ...address this..what? this is probably what you've mentioned above in a separate paragraph, you need to write briefly what exactly you mean

line 58-60: the meaning of sentence is interesting but the phrasing is poor

line 61: are the changes of the Subpolar gyre known or unknown within the 3 ka? this would justify the usage of "infer"

line 64: give location of the second site before giving info about the first site (Sentence line 61 states two sites). If the second site is used only for comparison, you have to rethink another way to announce your study sites

line 70-73: sentence far too long

line 75-76: "two independent proxies" which are...?

line 76: Following data obtained on plankton tow...

line 77: avoid "calculating" as it calls for a formula or an explanation

line 79-83: sentence too long

line 85-88: info about Nps should go up before d18O difference between the two species. Info about T. quinqueloba (water temperature preference, etc) should also be given

line 91: two complementary proxies....which are?

line 101: the polar front doesn't extend northward, it comes from the north, if it extends that would be southward, if the authors mean something else, please rephrase

line 102: "clear increases"?? maybe "clear" means marked...obvious...increases

line 103: sea-ice abundance?? rephrase

line 104: similar with eastern Labrador Sea? do you mean your data? if so, state it clearly

line 117: ...in LSW using marine sediment cores for paleoceanographic reconstructions is challenging due to the absence of suitable sites bathed by LSW waters.

line 120: again "infer" is a strong verb with a strong meaning

line 155: unless I'm mistaken, Site RAPiD-21-12B is mentioned no where on the map or in the text above!! further more the sentence where it is mentioned is unclear.

line 158: why the same? and same with what?

Comments on Figures:

Figure 1: please locate Site RAPiD-21-12B on the map.

Figure 3a: what does SPMW stands for?

Figure 3a: the color-coding of ϵNd data compared with $d13\text{C}$ is very similar and hardly distinguishable, please chose another color for either one.

Reviewer #3 (Remarks to the Author):

Moffa-Sanchez and Hall present 3000 years of high-resolution paleoceanographic proxy data from a core in the Labrador Sea. They interpret their data in terms of Labrador Sea Water production and subpolar gyre strength, and infer that both weakened during centennial-scale cold intervals including the Little Ice Age. This is an extension of the 1200 year record given in Moffa-Sanchez et al. (2014) with similar interpretation. While the longer record is interesting the interpretations with respect to LSW are still not clear, and the advance beyond the 2014 paper is not fundamental.

The authors preface their discussion on the observation that LSW formation depends on surface density and therefore surface salinity. They cite modern studies on the Great Salinity Anomaly that show reduced formation in association with freshwater input from the Arctic. A logical leap is then made to infer that the relative influence of the EGC (fresh) and Irminger Current (salty) should control LSW convection on centennial timescales. Critically there is no mention (at the start) of wind and thermal buoyancy, when in fact recent variations in LSW can be explained almost entirely through atmospheric forcing (NAO) driving evaporation and heat loss over the Labrador Sea (Yeager and Danabasoglu, 2014). The atmosphere (NAO) is raised toward the end of the paper but I don't find the arguments for dismissing it very compelling.

Two proxies are then introduced, one that is traditionally taken as a temperature indicator (%Nps) and the other meant to capture vertical thermal stratification. The stratification proxy is the $d18\text{O}$ difference between Nps and T. quinqueloba. A reference is made to Simstich et al. (2003) as the basis for this proxy, but the authors need to give a clear explanation of how this works. My

understanding of the Simstich study is that a large difference indicates Tq is living in warm and salty surface waters (with Nps in colder waters below), and a small difference means Tq is living in cold and fresh surface waters (with Nps in saltier waters below). Here the authors focus on the salinity signal, again at the expense of the temperature signal: the small difference during cold periods is taken as evidence of surface freshening (OK) and therefore weak LSW. But a small d18O gradient also implies a small density gradient, meaning that these data could be interpreted in the opposite sense: that during the LIA the surface cooling was more important than the freshening, and LSW formation consequently increased. It is the authors' counterintuitive use of a stratification proxy (smaller d18O gradient = more stratification) that is hard to get past. There may be an explanation, such as changing depth habitats, or d18O being a poor measure of density here, but it would need to be explained. The Tq d18O record shows a shift toward heavier values during the LIA consistent with surface cooling exceeding freshening, while the shift between 2000 and 1500 BP is dominated by an Nps (subsurface) d18O change. There may be some opportunity to decouple spring/summer temperature (season the proxies record) and winter temperature (season when convection occurs) but I suspect that is not so easy.

The story of this paper is essentially an extension of Moffa-Sanchez et al. (2014). Therefore the question of whether this paper is an "advance in understanding likely to influence thinking" depends in part on the longer temporal context. Here I find the comparisons to other records to be reasonable in that they show some plausible synchronicities between the LSW inferences and cooling, ice, SPG salinity, and ISOW speed. Given that these are all influenced by different factors, one should not expect very strong correspondence: they are what they are. However, mainly due to the continued ambiguity of the stratification proxy interpretation, I do not necessarily feel more persuaded about LSW formation and its relation to cold events in the North Atlantic. The d13C and epsilon-Nd records are too short to be useful.

One suggestion is to base the vertical gray bars on the data compiled in Figure 4, to make them independent of the data in Figures 2 and 3.

Reviewers' comments:

Reviewer #1 (Remarks to the Author):

This study presents new proxy data from two marine sediments cores, which are used to reconstruct variability in the North Atlantic Subpolar Gyre during the late Holocene. The authors link influx of Arctic waters and Labrador Sea Water formation to climatic conditions in Europe. The results are compared to other relevant published records from the region, providing convincing evidence for the conclusions presented in this manuscript. The study of climate variability of the last millennia and its link to ocean dynamics is crucial for understanding present and future changes and this study makes a significant contribution. I can therefore recommend the publication of this article and have only very minor suggestions for revisions.

Thank you.

I suggest visualizing the two main scenarios that are discussed in the paper by including a conceptual figure/map. European cold period with polar waters in the E Lab Sea and reduced LSW formation vs. the warm periods with contrasting conditions. Each scenario can be shown on a map with currents and front positions as well as a profile across the main sites showing the water masses.

We have now included a schematic figure as Figure 5 which highlights the general circulation schemes associated with both warm and cold conditions.

Line 42 Should refer to Figure 2e, not Figure 2d

Apologies, this typo has been corrected.

Line 288 “Unlabeled red circles” to indicate corals locations are not shown on the map?

Changed to diamonds.

Line 320 Figure 4: include the marine records as in Figure S2.

Added to Figure 4.

Reviewer #2 (Remarks to the Author):

Moffa-Sanchez and Hall present an interesting study on decadal-scale hydrographic changes in the subpolar gyre during the period of last 3000 years BP and formation changes of deep water production in the Labrador Sea compared with Iceland-Scotland Overflow Water (ISOW) using planktonic foraminifer stable isotopes and sortable silt data.

The study is certainly of interest to improve the very little knowledge of the formation of Labrador Sea deep water and the strength of the northwest boundary of the subpolar gyre, yet the results could hardly reconstruct the behavior of the gyre at its center and consequences in its northeast boundary (along the path of the North Atlantic Drift).

We appreciate the reviewer acknowledges that the paper contributes to our understanding of LSW formation and strength of the northwest boundary of subpolar gyre. We also understand the reviewers concerns regarding the need for additional proxy reconstructions to form a spatial network across the wider subpolar gyre, but at present we are largely

constrained by the lack of appropriate sites in the centre of the subpolar gyre/LSW that would allow high-resolution proxy reconstructions of ocean conditions to be generated and this still remains a challenge. In this study, we attempt to fill the gap the knowledge by focussing on variability in LSW formation and subpolar gyre dynamics, by utilizing new and published proxy records which albeit indirectly, can collectively be used as indicators of past changes in the formation of LSW and SPG dynamics. These also include the eastern component of the SPG from Thornalley et al., (2009).

Although, the study finds some links between major millennial-scale cold events within the last 3 ka and major historical events, the discussion at a decadal-scale of those links is somehow unsatisfactory.

We made a conscious decision to not discuss the decadal events since the dating uncertainties for decadal short-lived events found in the records would be too large to compare across cores and discuss robustly. Additionally, centennial-scale variability in the records presented shows the most coherent variability across the records (particularly when comparing to the published lower temporal resolution records which is in our opinion a strong finding in this manuscript).

The organization of the manuscript, presentation of the results and the general discussion need some improvements and revisions.

Specific comments on the manuscript:

I find the introductory paragraph very general, lacking specific characteristics of the study region. However, the reader could find explanatory sentences of specific oceanic setting along the manuscript, right before some results or discussions. The authors should consider moving those sentences to the intro paragraph.

As an example, lines 124-129, lines 145-153.

We discuss the individual ocean settings just before presenting each proxy record to have a flow in the manuscript and to avoid the reader referring back to the ocean setting paragraph at the beginning each time a new proxy record is introduced. This structure was chosen as the nature of some of the proxies presented require a thorough discussion. This was done to ease the reader since Nature Communications has a broad readership and to have a logical narrative in the manuscript. We have followed the reviewer's advice and we have now introduced the two new marine sediment cores and briefly described their ocean setting which we then describe in detailed at a later stage in the manuscript (Line 66-80).

The records show far more variations (at a decadal scale) than the ones the authors refer to by the grey thick lines (figures 2 and 3): The decrease of Nps to 60-40% is brief, the following intervals of increase are quicker even within the intervals that are identified here as being intervals of enhanced influence of warm water. how could the authors explain this? are there more variations than the ones the authors identify? are there variations within the main variations (more frequent frontal migrations)?

We are aware that there is considerable decadal variability in the records, however because of the reasons mentioned above we tried to avoid discussing decadal scale variability and focused only on centennial time-scales. The shaded bars refer to the centennial scale periods of glacial advances and retreats that were established by Denton and Karlen (1973) and later

used by Mayewski (2004) in the Holocene centennial climate events review. We use this same temporal framework and the grey bars help to highlight the good correspondence with our North Atlantic records. However, in order to address the reviewer's concerns and ease the reader to identify the above and below-average events at all time scales, we have added to Figure 3 an average line across the records.

Looking at the location of the Site RAPid-12-1K, referred to as south of Iceland is a bit tricky and somehow misleading. The Site is located in the main path of the North Atlantic Drift, southeast of Iceland.

We have corrected this (line 76).

The “behavior” of the subpolar gyre from the “perspective” of the site location is true (line 131-134), yet the comparison with the Study site raises some concerns. Even though both set of results show a clear similar trend, the S-E Labrador Sea results clearly tell another story embedded within the one that the S-E Iceland results are telling. The movements of the Polar-Arctic fronts are clearly more frequent south of the Labrador Sea than S-E Iceland where warm and salty water flows are much stronger. Is there an influence on the recovery/shut down of LSW formation?

We agree with the reviewer that the two ocean frontal systems (SW Greenland and SE Iceland) for the Polar (PF) and Subpolar front (SPF), respectively are very different in nature. The PF is a very strong front and its southern advancement can be more sensitive to changes in the volume of the southward flowing EGC. Conversely, the SPF is thought to be responding to the mixing between subpolar versus subtropical waters driven by frontal shifts making the changes more subtle and also being driven by the gyre dynamics. Yet, it is very possible that the low sedimentation rates of RAPid-12-1K (the data from the SPF by Thornalley et al., (2009) resulted in a smoother record not capturing the multidecadal variability and hence it is not possible to assess the influence on the recovery/shutdown of LSW formation or the different dynamics of these two fronts robustly within the contexts of the currently available data.

Although the results of the second study site (21-3K) improve the discussion about deep water formation (ISOW) and the strength of the subpolar gyre (Salinity variations), they come in the manuscript far too late, I wonder if it's not best for the manuscript to introduce both sites at the same time, discuss the results and compare with other results at the same time, that would be more concise and clearer.

The Sortable Silt results are, in my opinion, not given enough attention as they represent a crucial factor in the discussion of ISOW flow speed and contribution of LSW, two or three sentences would be helpful to fully cover the results acquired.

Yes, we agree with the reviewer and we have brought the SS record forward in the manuscript and highlighted the importance of this record, which will clarify the confusion about the amount of new data presented in this manuscript (Reviewer#3). SS results are now found from Line 165. We have added some discussion about the absolute change in bottom flow speeds in light of the recent paper in press by McCave et al., (line 171).

Furthermore, how did the authors deal with the influence of Ice rafted debris (direct sedimentation not representative of bottom sorting)? Knowing that the study Site is close by

Bond's MC52 site where influence of IRD has been recorded. I suggest having a look at the work of Hass, H. C. (2002), A method to reduce the influence of ice-rafted debris on a grain size record from northern Fram Strait, Arctic Ocean. *Polar Research*, 21: 299–306. doi:10.1111/j.1751-8369.2002.tb00084.x

The reviewer raises an important issue with SS measurements. We acknowledge that some criticism of the SS approach has focussed on the potential that the sediment properties are mainly derived from source. However, we maintain that the current imposes a sorted signature whose properties are a function of flow speed (c.f. McCave and Hall 2006; McCave et al., in press). Several studies have highlighted the potential IRD influence on SS measurements in glacial NE Atlantic sediments. However, these are only during glacial conditions, and hence due to the recent timescales of our study the IRD reaching the site would be expected to have been minimal and within the competency of the ISOW flow to sort the sediment. To a large extent the evidence or signature for that suggestion is in the SS vs SS% plot of many similar SS records (including glacial Gardar Drift sediments) where current-sorted trends are apparent and dominate the SS signal (c.f. McCave and Hall 2006). Unfortunately, we don't have SS% data as our SS measurements have been made with a Coulter Counter. Hass (2002) use the %sand in order to correct for IRD input, the argument is that the %sand will not be easily moved by the bottom current and will be affected by the flux of IRD. We compared the % sand (including biogenic carbonate) to the SS measurements and find no correlation between these two (Figure 1). We therefore concluded that the amount of IRD in this core over the last 3000 years would be minimal and we decided not to correct for it. This is consistent with the small (~1.6 μm) range of SS suggesting flow speed variability on the order of ~2.2 cm/s according to the calibration recently presented in McCave et al., in press Deep Research Part I. Furthermore, we are confident on the minimal effect of IRD at this site since the SS record presented by Boesenkool et al., 2007 (the companion box-core from the same site) which spans the last 230 years and hence includes the instrumental record shows similar grain sizes.

Figure R1. Sortable Silt Mean Grain Size and %Sand from RAPiD-21-3K.

The conclusion is drawn from the results and the discussion includes comparison with previous records. However, I'm not keen about the relevance of the general discussion involving the historical evolution of the European societies and the climatic influence on them. Two pages have been dedicated to the later discussion, setting apart the discussion of the climatic context (or mentioned rarely) that have been reconstructed and discussed above. I personally like those kinds of cross-examinations of Human evolution built on knowledge of environmental changes. Should the authors look for a more included discussion while discussing the study's results rather than leaving it to the very end? the answer is probably a matter of opinion.

We make the linkage between the ocean variability and European climate via the comparison to terrestrial and historical records. The historical evidence compiled in Figure 4 is used as additional evidence of the implications that the ocean variability may have had on several events recorded in European history very nicely illustrating the ocean-land linkage. We have not spent two pages discussing the climate influence on European societies, we purposefully limited the discussion to line 235-249 and Figure 4 since this topic is partly tangential to the main thrust of the paper, but nonetheless of interest to the broad readership of Nature Comms.

I suggest that figure 4 display the events (red and blue) along with the variations of Nps and d18O rather than with the three grey thick lines.

We have now added the marine records from the LSW/SPG to the bottom of Figure 4 to aid comparison.

I am listing below my comments in order of the manuscript and not scientific weight.

line 42: correct Figure 2e instead of Figure 2d, referring to Ref 14

Corrected.

line 44: ...there are very limited data...

Corrected.

line 44: "confirm" is a strong verb; no need to add fresh before ice!; the sub polar gyre needs to be introduced beforehand

Replaced by support.

line 44-47: the whole sentence needs to be rewritten and broken down in many sentences if possible.

Rewritten.

line 47-49: redundancy: the lack of data has already been mentioned at the beginning of line 44

We understand this point but we also feel that the reason why there is a lack of data from this region should be explained and this is the information added in the second sentence.

line 50: ...address this..what? this is probably what you've mentioned above in a separate paragraph, you need to write briefly what exactly you mean

Changed.

line 58-60: the meaning of sentence is interesting but the phrasing is poor

Corrected.

line 61: are the changes of the Subpolar gyre known or unknown within the 3 ka? this would justify the usage of “infer”

From the different proxy records some conclusions are drawn and therefore the use of infer may be appropriate here.

line 64: give location of the second site before giving info about the first site (Sentence line 61 states two sites). If the second site is used only for comparison, you have to rethink another way to announce your study sites

We have now modified by restructuring and reordering the main text.

line 70-73: sentence far too long

Shortened.

line 75-76: “two independent proxies” which are...?

Changed.

line 76: Following data obtained on plankton tow...

Corrected.

line 77: avoid “calculating” as it calls for a formula or an explanation

Replaced by taking.

line 79-83: sentence too long

Shortened.

line 85-88: info about Nps should go up before d18O difference between the two species. Info about T. quinqueloba (water temperature preference, etc) should also be given

This section has been modified we hope it addresses this.

line 91: two complementary proxies....which are?

Added.

line 101: the polar front doesn't extend northward, it comes from the north, if it extends that would be southward, if the authors mean something else, please rephrase

Rephrased.

line 102: “clear increases”?? maybe “clear” means marked...obvious...increases

Replaced by marked.

line 103: sea-ice abundance?? Rephrase

Corrected.

line 104: similar with eastern Labrador Sea? do you mean your data? if so, state it clearly

Corrected.

line 117: ...in LSW using marine sediment cores for paleoceanographic reconstructions is challenging due to the absence of suitable sites bathed by LSW waters.

Rephrased.

line 120: again “infer” is a strong verb with a strong meaning

We use the verb ‘infer’ to mean deduce or conclude (something) from evidence and reasoning rather than from explicit statements. So it is in this case justified.

line 155: unless I’m mistaken, Site RAPiD-21-12B is mentioned no where on the map or in the text above!! further more the sentence where it is mentioned is unclear.

Modified.

line 158: why the same? and same with what?

Same as ref. 35. Modified.

Comments on Figures:

Figure 1: please locate Site RAPiD-21-12B on the map.

To avoid a cluttered figure we have clarified that these two cores are from the same site in the main text (l.158).

Figure 3a: what does SPMW stands for?

We have added this into the figure caption (l. 344).

Figure 3a: the color-coding of ϵNd data compared with $\delta^{13}\text{C}$ is very similar and hardly distinguishable, please chose another color for either one.

Apologies, we have now changed the colour of the $\delta^{13}\text{C}$ to brown, we hope this is more distinguishable.

Reviewer #3 (Remarks to the Author):

Moffa-Sanchez and Hall present 3000 years of high-resolution paleoceanographic proxy data from a core in the Labrador Sea. They interpret their data in terms of Labrador Sea Water production and subpolar gyre strength, and infer that both weakened during centennial-scale cold intervals including the Little Ice Age. This is an extension of the 1200 year record given in Moffa-Sanchez et al. (2014) with similar interpretation. While the longer record is interesting the interpretations with respect to LSW are still not clear, and the advance beyond the 2014 paper is not fundamental.

As the reviewer states we present an extension of the record presented in Moffa-Sanchez (2014). However, we also present a new sortable silt record from RAPiD-21-3K spanning the last 3000 years. This record is at an unprecedented temporal resolution (subdecadal) for the last ~3000 years and this new data confirms the impact of changing freshwater inputs on

Labrador Sea Water formation via the modification of the intermediate waters in the Iceland Basin. Therefore the new data in this manuscript of sufficient difference to the Moffa-Sanchez (2014)). The coherency amongst the records and the terrestrial and historical records across well-known climatic events adds value to this manuscript. We apologise for the confusion and we have now tried to make more explicit the novelty of this dataset by discussing this data in more detail earlier in the manuscript as suggested by reviewer#2.

The authors preface their discussion on the observation that LSW formation depends on surface density and therefore surface salinity. They cite modern studies on the Great Salinity Anomaly that show reduced formation in association with freshwater input from the Arctic. A logical leap is then made to infer that the relative influence of the EGC (fresh) and Irminger Current (salty) should control LSW convection on centennial timescales. Critically there is no mention (at the start) of wind and thermal buoyancy, when in fact recent variations in LSW can be explained almost entirely through atmospheric forcing (NAO) driving evaporation and heat loss over the Labrador Sea (Yeager and Danabasoglu, 2014). The atmosphere (NAO) is raised toward the end of the paper but I don't find the arguments for dismissing it very compelling.

We agree with the reviewer and we have added a mention to the wind stress component to the introduction of the manuscript (Line 35).

In our opinion paragraph starting Line 250 is a very measured discussion about past NAO reconstructions. We cite all the available work that have attempted to reconstruct the atmospheric conditions over the North Atlantic during this period and conclude that according to some records the atmospheric circulation potentially also changed in phase with the ocean records. However, we also add three cautionary arguments which are: (i) that NAO proxy reconstructions based on single or two proxy reconstructions have shown to not be very robust especially when compared to analysing big datasets (Ortega et al., 2015) likely because of the challenges associated reconstructing SLP patterns that may not have remained stationary, (ii) the fact that proxy records show coherent variability with the ocean at centennial time-scales could be a response to the ocean conditions and not necessarily a driver and (iii) recent model studies suggest the secondary role that wind forcing can have on SPG/LSW changes at centennial time-scales.

Due to the lack of robust atmospheric reconstructions, at this stage it is not possible for us to make a stronger case for the implications of atmospheric circulation and hence we focus mostly on the oceanographic component which is our contribution from our data.

We have however rephrased this section for clarification (Line 262-279).

Two proxies are then introduced, one that is traditionally taken as a temperature indicator (%Nps) and the other meant to capture vertical thermal stratification. The stratification proxy is the d18O difference between Nps and T. quinqueloba. A reference is made to Simstich et al. (2003) as the basis for this proxy, but the authors need to give a clear explanation of how this works. My understanding of the Simstich study is that a large difference indicates Tq is living in warm and salty surface waters (with Nps in colder waters below), and a small difference means Tq is living in cold and fresh surface waters (with Nps in saltier waters below). Here the authors focus on the salinity signal, again at the expense of the temperature signal: the small difference during cold periods is taken as evidence of surface freshening

(OK) and therefore weak LSW. But a small d18O gradient also implies a small density gradient, meaning that these data could be interpreted in the opposite sense: that during the LIA the surface cooling was more important than the freshening, and LSW formation consequently increased. It is the authors' counterintuitive use of a stratification proxy (smaller d18O gradient = more stratification) that is hard to get past. There may be an explanation, such as changing depth habitats, or d18O being a poor measure of density here, but it would need to be explained.

We appreciate the reviewers concerns about this proxy and the counterintuitive nomenclature of upper ocean stratification stratification. We try to address the reviewers comments in two parts: (1) $\Delta \delta^{18}\text{O}_{\text{Nps-Tq}}$: an indicator of the presence of Atlantic Waters including a detailed explanation of the proxy its interpretation, and (2) Additional lines of evidence that support our interpretation of this proxy.

1. $\Delta \delta^{18}\text{O}_{\text{Nps-Tq}}$: an indicator of the presence of Atlantic Waters

1.1. Application of $\Delta \delta^{18}\text{O}_{\text{Nps-Tq}}$

*We interpret the difference in the $\delta^{18}\text{O}$ composition of these two foraminifera species following the study by Simstich et al. (2003), in which they present $\delta^{18}\text{O}$ data from these two species from a number of core-tops (which they also contrast with plankton tow and sediment trap data) across the Nordic Seas from Greenland to Norway. They find the near-surface calcification depths of *T. quinqueloba* (25-75m) but observe changes in the calcification depth of *N. pachyderma* (Nps) across this longitudinal transect (Figure 2). Under the presence of warm Atlantic Waters Nps calcifies deeper in the water column likely due to its preference for cold waters (e.g. Eynaud (2011) and references therein). However in the East under the EGC both species calcify at the surface. It has been repeatedly proposed that Nps is able to change its calcification depth depending on the hydrography with a tendency to follow same density lines and hence the regional hydrography (Hillaire-Marcel et al., 2001; Moffa-Sánchez et al., 2014; Simstich et al., 2003). Across the Nordic Seas the depth of Nps follow the temperature on the west under Atlantic Waters and the salinity on the East (EGC). We find the most compelling evidence in Figure 7 from Simstich et al. (2003) (See below) which shows the spatial distribution of the difference between the two species $\delta^{18}\text{O}$ across the front between the Irminger Current and the East Greenland Current in the Irminger Sea.*

Fig. 7. Spatial distribution pattern of $\delta^{18}\text{O}$ differences between *Neoglobobulimina pachyderma* (s) and *Turborotalita quinqueloba*. High values delineate the inflow of Atlantic water into the Norwegian and Irminger Seas. Low values document influence of low-saline surface water of polar origin. Large numbers label $\Delta\delta^{18}\text{O}$ isolines.

This is what is likely also the case further south at RAPID-35-COM. We are therefore confident that the difference between these two species is not only governed by the Tq (that sits in the surface and can change considerably) but despite Nps following the same density line the subtraction yields more confidence since the $\delta^{18}\text{O}$ of Nps in the Nordic Seas was measured to change up to 1per mil between these two regimes (Figure R2).

Figure R2. $\delta^{18}\text{O}$ values for calcite are calculated and plotted as lines. The Nps are shown in grey circles and the Tq in stars. This figure clearly shows the depth variability of Nps according to the watermasses present in the upper water column. The EGC waters are characterised by a smaller difference of the $\delta^{18}\text{O}$ between the species and a larger in the Norwegian Current. From Simistich et al. (2003).

We have modified the discussion and explanation of this proxy in the main text (1.96-121).

1.2. Decrease in thermal stratification: Southward shift of polar front or increase upper ocean mixing due to Labrador Sea Water formation?

The reviewer questions that the decrease in 'thermal stratification' could be interpreted as better mixed upper water column and therefore an increase in the formation of Labrador Sea Waters as opposed to more polar waters reaching the site due to a southward shift of the polar front. We understand the potential ambiguity of this proxy but there are several arguments and lines of evidence that additionally strengthen our interpretation of the proxy

results. (Note that we have deleted the mention of thermal stratification to avoid confusion in the main text and we have also included a more comprehensive explanation of this proxy).

Firstly, the blooming time of the foraminifera is spring/summer (e.g. Jonkers et al. 10), at this time the surface of the Labrador Sea is undergoing restratification from its boundary currents from the northeast due to a weakening of the westerlies and addition of the meltwater from Greenland and sea-ice driven by the increase in solar radiation (Lilly and Rhines, 2002; Straneo et al., 2002) and shown in hydrographic transects taken in the spring/summer across the Labrador Sea (Holliday et al., 2009; Yashayaev, 2007). Therefore, the eastern location of RAPiD-35-COM and the blooming season of the foraminifera measured means that these enable monitoring mainly the summer restratification process rather than the winter convection. We have strengthened the summer restratification process in the main text (line 84-95).

Secondly, the large amplitude of variability of the %Nps record (~50%) would be equivalent to a 4-6°C (e.g. Eynaud, 2011 and references therein) change of the surface waters across the last 3kyr. This large temperature variability would not be possible to be explained by just vertical mixing from deepwater formation events in the Labrador Sea.

Thirdly, we agree with the reviewer that we can only reconstruct temperature conditions and we argue for salinity to play a bigger role in the preconditioning for convection. However, here we present the temperature and salinity hydrographic data between 1950-2013 of the top 150 (taken mostly in spring/summer) (courtesy of I. Yashayaev) (Figure R3) and conclude that over the last 60 years the temperature and salinity in the surface Labrador Sea present a positive relationship ($R=0.63$ annually, $R=0.83$ decadal, Figure R4). This relationship nicely demonstrates that at multidecadal time-scales our reconstructed changes in temperature will likely be accompanied by salinity changes.

Figure R3. Temperature and Salinity time-series from the top 150m in the Labrador Sea comprising AR7 and Argo (Courtesy of I. Yashayaev). The data is presented as anomalies from the mean.

Figure R4. Annual (top) and 9-point running mean (bottom) temperature and salinity data from the time-series presented in Figure R3.

In order, to clarify the temperature and salinity relationship, this information has now been added to the Supplementary Material and a few sentences have been added to the manuscript (Line 116).

2. Additional lines of evidence supporting our proxy interpretation:

2.1. Comparison to other proxy records around Greenland:

There are a growing number of ocean records from around Greenland and the Arctic. Yet, the lower temporal resolution and dating uncertainties make the comparison between records challenging. We have attempted to thoroughly review and summarise the available data across different geographical locations and conclude that common ocean conditions are recorded in several records and are broadly consistent with our findings and support the mechanisms proposed in the manuscript, strengthening our interpretation for an increase in the proportion of polar waters reaching the E Labrador Sea at the cold times recorded in the %Nps and during small $\Delta \delta^{18}O_{Nps-Tq}$.

*The **Fram Strait** is a key location to monitor exchanges between the Arctic and the Nordic Seas (norward flowing Atlantic Waters and southward flowing Arctic EGC waters). IP25 sea-ice reconstructions reveal a Neoglaciation increase in sea-ice with a concomitant increase in drift ice (IRD) near Fram Strait around 3kyrs (Müller et al., 2012) paralleled to a millennial increase in Arctic waters indicated by foraminiferal assemblage counts (Werner et al., 2014) and radiogenic isotopes in sediments indicate a southeastern shift of the marginal sea ice zone from 2800yrs BP (Werner et al., 2016). The sea ice record shows centennial increases centred ~2800, 2300, 1600 and 500yrs BP (Müller et al., 2012).*

*Further downstream of the Fram Strait, along the **East Greenland coast** several paleocenaographic records indicate fluctuations in drift ice and sea ice, foraminiferal representative of subsurface water conditions reveal stronger Atlantic intermediate water influence between 1.4-2kyrs with a short warming centred around 900yrs BP (Perner et al., 2015). Sea-ice reconstructions from the same core reveal minimum sea ice between before 1.4kyrs with increase at 1200yrs BP and a minimum at 1000yrs BP and highest values 700-0yrs BP (Kolling et al., 2017).*

*Further downstream around the **Denmark Strait**, lower resolution records have found increase in the influence of EGC and a reduced influence of IC waters from 4kyrs BP using the percentage of an Atlantic species *C. neotrretis* and interpreted as an advance of the polar front during the Neoglaciation (Jennings et al., 2011) and a similar increase in IRD reaching North of Iceland and the East Greenland Shelf (Andrews et al., 2010; Moros et al., 2006a; Perner et al., 2016). Similarly to the records from East Greenland (Perner et al., 2015), higher resolution records show a broad increase in the % of IC foraminiferal species between 1.5-2.7kyrs BP paralleled by an increase in productivity, which is also seen in the % of Atlantic species of coccoliths North of Iceland (Andrews and Giraudeau, 2003) (See figure 4 from Perner 2016). Similarly, low-resolution carbonate flux and % *N. teretis* records, indicate warm periods centred at 2kyrs and 1100-700yrs AD and cold at 2.6kyrs (Jennings et al., 2002) and during the LIA (Jennings and Weiner, 1996). Additionally, IRD records and quartz/pyroxene records from the Denmark Strait present high numbers at 2.5kyrs and 0.5kyrs with low around 1.1-1.5kyrs (Perner et al. 2016). **North Iceland** sea ice, IRD and temperature records are also consistent with some of these centennial events (Cabedo-Sanz et*

al., 2016; Jiang et al., 2015; Massé et al., 2008; Moros et al., 2006a) (Figure 2 of manuscript).

Records from **South Greenland** have a low temporal resolution and the results remain a bit more equivocal largely because of the coastal location of the sediment cores. Whilst Andresen et al. (2013) shows low salinity and sea ice increase between 1500-700 years BP although Jensen et al., (2004) shows an increase in EGC influence at 1300yrs AD at the onset of the LIA with a warm period from 500-1300 yrs AD based on diatoms. Yet, another foraminiferal assemblage record does not present much variability either from the same site (Lassen et al., 2004). Similarly, Miettinen et al., (2015) records a shift in temperatures and sea-ice at the onset of the LIA although the early part of the record doesn't present any features. Additionally, IRD reconstructions from a nearby site in the Eirik Ridge by Alonso-Garcia et al. (2016) reveals an increase in the Hematite Stained Grains (with origin from NE Greenland) during the LIA, confirming the increase in polar waters during the LIA and hence perhaps analogously to other cold events. However, Seidenkrantz et al., (2007) present some conflicting results but this record from SW Greenland represents oxygenation conditions in the fjord and it is therefore hard to interpret as the WGC.

Records further North on the **West Greenland** the area around Disco Bugt is influenced by the West Greenland Current is a mixture of EGC and IC, therefore ocean reconstructions from here can be indicative of the proportions further south and confirm our data from RAPID-35-COM. The data from Disco Bugt show broadly a Neoglacial increase in the influence of polar waters punctuated by distinct centennial events consistent with our records from SE Greenland. Sea ice reconstructions from diatoms reveal cold periods at 2700yrs and 700-0 years BP with a warmer interval with more Atlantic waters in the WGC around 2200-1500yrs BP (Moros et al., 2006b). Sea ice diatom reconstructions also show warm-water species reduced sea ice 1900-1500 years BP coincident with the RWP (Sha et al., 2014) also present in the foraminiferal assemblage reconstructions that reveal an increase in Atlantic versus Polar waters between ~1500-2100 yrs BP (Perner et al., 2011; Seidenkrantz et al., 2008). Another obvious feature from these records is the LIA cooling. Additional low resolution reconstructions from the **Canadian Arctic** reveal low sea ice and less bowhead whales in Canadian Arctic Archipelago 1.5-0.8kyrs BP and higher after 0.8kyrs BP and 3-1.5kyrs BP (Belt et al., 2010), consistent with some of the periods seen on West Greenland.

The conclusion from this exhaustive review of the available data is that indeed an increase in the influence of EGC waters from 3kyrs to present is present around Greenland. However, the centennial scale changes in the southward export of EGC waters is more of a challenge largely because of the limited datasets with a decadal resolution from some of these regions but some of these centennial events are seen in a large number of the records as summarized above and largely coincide with our records from SW Greenland.

A shortened version of this review has been included in the Supplementary Material (section 3).

2.2. Corroboration of the mechanisms by physics-based model studies:

A new model study by Moreno-Chamarro et al., 16 shows the sensitivity of the gyre and Labrador Sea Water to influx of Arctic freshwater during the LIA. The model results show that the increase in the Arctic waters to the subpolar North Atlantic can trigger the cold

events such as the LIA. We provide in this manuscript a compilation of data, which provides unique evidence for this mechanism and our data for other centennial scale events show very similar behaviour.

The Tq d18O record shows a shift toward heavier values during the LIA consistent with surface cooling exceeding freshening, while the shift between 2000 and 1500 BP is dominated by an Nps (subsurface) d18O change. There may be some opportunity to decouple spring/summer temperature (season the proxies record) and winter temperature (season when convection occurs) but I suspect that is not so easy.

We state in Line 185 about the potential for seasonality changes in the millennial scale variability. The shifts in the centennial variability would not be very straightforward and may induce more confusion.

The story of this paper is essentially an extension of Moffa-Sanchez et al. (2014). Therefore the question of whether this paper is an “advance in understanding likely to influence thinking” depends in part on the longer temporal context.

We hope we have clarified this in the previous reply with the new SS record from RAPID-21-3K (Figure 3d). The mechanisms proposed in Moffa-Sanchez et al., 2014 still hold but here we clearly show with the new SS record the effects that the freshwater input has on the intermediate depths via changes in LSW formation. Additionally, this mechanism holds for the rest of the climatic oscillations with similar timing to the variability observed in NW Europe, an ocean-land linkage that has been absent from the literature due to the lower resolution marine records.

Here I find the comparisons to other records to be reasonable in that they show some plausible synchronicities between the LSW inferences and cooling, ice, SPG salinity, and ISOW speed. Given that these are all influenced by different factors, one should not expect very strong correspondence: they are what they are.

We agree with the reviewer that the sediment cores are far apart and each of them will be influenced by very different processes. However, these records are all also related indirectly to SPG/ LSW dynamics and it is hence very possible that their coherent signals are due to the processes happening within the SPG/LSW region and hence strengthen our conclusions.

However, mainly due to the continued ambiguity of the stratification proxy interpretation, I do not necessarily feel more persuaded about LSW formation and its relation to cold events in the North Atlantic. The d13C and epsilon-Nd records are too short to be useful.

We hope we have clarified some aspects of the stratification proxy ambiguity above and have strengthen the co-variability between the sortable silt record and the PF shifts SE Greenland. The two low resolution chemical signature records (Copard et al., 2012; Oppo et al., 2003) are shorter and don't fully span the entirety of the last 3000 years, but when combined they do and the centennial scale variability looks very similar to the subpolar front changes shown by Thornalley et al 2009 and also with the new SS record presented here. We therefore find these two additional proxy carriers and datasets very convincing in the context of the new and published datasets.

One suggestion is to base the vertical gray bars on the data compiled in Figure 4, to make them independent of the data in Figures 2 and 3.

The grey bars are consistent through the manuscript and they indicate the periods that have been used as a framework in the paleoclimatic literature for this time period. The time periods are based on the advance and retreat of the glaciers around the North Atlantic from the records by Denton and Karlen et al., 1973, these are used in Holocene event review by Mayewski et al., (2004) as a temporal framework.

References:

!!! INVALID CITATION !!! .

Alonso-Garcia, M., Kleiven, H.F., McManus, J.F., Moffa-Sanchez, P., Broecker, W., Flower, B.P., 2016. Freshening of the Labrador Sea as a trigger for Little Ice Age development. *Clim. Past Discuss.* 2016, 1-30.

Andresen, C., Hansen, M., Seidenkrantz, M.-S., Jennings, A., Knudsen, M., Nørgaard-Pedersen, N., Larsen, N., Kuijpers, A., Pearce, C., 2013. Mid-to late-Holocene oceanographic variability on the Southeast Greenland shelf. *The Holocene* 23, 167-178.

Andrews, J.T., Giraudeau, J., 2003. Multi-proxy records showing significant Holocene environmental variability: the inner N. Iceland shelf (Hunafloi). *Quaternary Science Reviews* 22, 175-193.

Andrews, J.T., Jennings, A.E., Coleman, G.C., Eberl, D.D., 2010. Holocene variations in mineral and grain-size composition along the East Greenland glaciated margin (ca 67°-70°N): Local versus long-distance sediment transport. *Quaternary Science Reviews* 29, 2619-2632.

Belt, S.T., Vare, L.L., Massé, G., Manners, H.R., Price, J.C., MacLachlan, S.E., Andrews, J.T., Schmidt, S., 2010. Striking similarities in temporal changes to spring sea ice occurrence across the central Canadian Arctic Archipelago over the last 7000 years. *Quaternary Science Reviews* 29, 3489-3504.

Cabedo-Sanz, P., Belt, S.T., Jennings, A.E., Andrews, J.T., Geirsdóttir, Á., 2016. Variability in drift ice export from the Arctic Ocean to the North Icelandic Shelf over the last 8000 years: A multi-proxy evaluation. *Quaternary Science Reviews* 146, 99-115.

Copard, K., Colin, C., Henderson, G., Scholten, J., Douville, E., Sicre, M.-A., Frank, N., 2012. Late Holocene intermediate water variability in the northeastern Atlantic as recorded by deep-sea corals. *Earth and Planetary Science Letters* 313, 34-44.

Denton, G.H., Karlen, W., 1973. Holocene Climatic Variations--Their Pattern and Possible Cause. *Quaternary Research* 3, 155-205.

Eynaud, F., 2011. Planktonic foraminifera in the arctic: Potentials and issues regarding modern and quaternary populations, 1 ed.

Hass, H.C., 2002. A method to reduce the influence of ice-rafted debris on a grain size record from northern Fram Strait, Arctic Ocean. *Polar Research* 21, 299-306.

Hillaire-Marcel, C., De Vernal, A., Bilodeau, G., Weaver, A.J., 2001. Absence of deep-water formation in the Labrador Sea during the last interglacial period. *Nature* 410, 1073-1077.

Holliday, N.P., Bacon, S., Allen, J., McDonagh, E.L., 2009. Circulation and transport in the western boundary currents at Cape Farewell, Greenland. *Journal of Physical Oceanography* 39, 1854-1870.

Jennings, A., Andrews, J., Wilson, L., 2011. Holocene environmental evolution of the SE Greenland Shelf North and South of the Denmark Strait: Irminger and East Greenland current interactions. *Quaternary Science Reviews* 30, 980-998.

Jennings, A.E., Knudsen, K.L., Hald, M., Hansen, C.V., Andrews, J.T., 2002. A mid-Holocene shift in Arctic sea-ice variability on the East Greenland Shelf. *Holocene* 12, 49-58.

Jennings, A.E., Weiner, N.J., 1996. Environmental change in eastern Greenland during the last 1300 years: evidence from foraminifera and lithofacies in Nansen Fjord, 68°N. *The Holocene* 6, 179-191.

Jensen, K.G., Kuijpers, A., Koç, N., Heinemeier, J., 2004. Diatom evidence of hydrographic changes and ice conditions in Igaliku Fjord, South Greenland, during the past 1500 years. *The Holocene* 14, 152-164.

Jiang, H., Muscheler, R., Björck, S., Seidenkrantz, M.-S., Olsen, J., Sha, L., Sjolte, J., Eiríksson, J., Ran, L., Knudsen, K.-L., 2015. Solar forcing of Holocene summer sea-surface temperatures in the northern North Atlantic. *Geology* 43, 203-206.

Kolling, H.M., Stein, R., Fahl, K., Perner, K., Moros, M., 2017. Short-term variability in late Holocene sea ice cover on the East Greenland Shelf and its driving mechanisms. *Palaeogeography, Palaeoclimatology, Palaeoecology*.

Lassen, S.J., Kuijpers, A., Kunzendorf, H., Hoffmann-Wieck, G., Mikkelsen, N., Konradi, P., 2004. Late-Holocene Atlantic bottom-water variability in Igaliku Fjord, South Greenland, reconstructed from foraminifera faunas. *The Holocene* 14, 165-171.

Lilly, J.M., Rhines, P.B., 2002. Coherent eddies in the Labrador Sea observed from a mooring. *Journal of Physical Oceanography* 32, 585-598.

Massé, G., Rowland, S.J., Sicre, M.A., Jacob, J., Jansen, E., Belt, S.T., 2008. Abrupt climate changes for Iceland during the last millennium: Evidence from high resolution sea ice reconstructions. *Earth and Planetary Science Letters* 269, 564-568.

Mayewski, P.A., Rohling, E.E., Curt Stager, J., Karlén, W., Maasch, K.A., David Meeker, L., Meyerson, E.A., Gasse, F., van Kreveland, S., Holmgren, K., Lee-Thorp, J., Rosqvist, G., Rack, F., Staubwasser, M., Schneider, R.R., Steig, E.J., 2004. Holocene climate variability. *Quaternary Research* 62, 243-255.

Miettinen, A., Divine, D.V., Husum, K., Koç, N., Jennings, A., 2015. Exceptional ocean surface conditions on the SE Greenland shelf during the Medieval Climate Anomaly. *Paleoceanography* 30, 1657-1674.

Moffa-Sánchez, P., Hall, I.R., Barker, S., Thornalley, D.J., Yashayaev, I., 2014. Surface changes in the eastern Labrador Sea around the onset of the Little Ice Age. *Paleoceanography* 29, 160-175.

Moros, M., Andrews, J.T., Eberl, D.D., Jansen, E., 2006a. Holocene history of drift ice in the northern North Atlantic: Evidence for different spatial and temporal modes. *Paleoceanography* 21.

Moros, M., Jensen, K.G., Kuijpers, A., 2006b. Mid-to late-Holocene hydrological and climatic variability in Disko Bugt, central West Greenland. *The Holocene* 16, 357-367.

Müller, J., Werner, K., Stein, R., Fahl, K., Moros, M., Jansen, E., 2012. Holocene cooling culminates in sea ice oscillations in Fram Strait. *Quaternary Science Reviews* 47, 1-14.

Oppo, D.W., McManus, J.F., Cullen, J.L., 2003. Palaeo-oceanography: Deepwater variability in the Holocene epoch. *Nature* 422, 277-277.

Ortega, P., Lehner, F., Swingedouw, D., Masson-Delmotte, V., Raible, C.C., Casado, M., Yiou, P., 2015. A model-tested North Atlantic Oscillation reconstruction for the past millennium. *Nature* 523, 71-74.

Perner, K., Jennings, A.E., Moros, M., Andrews, J.T., Wacker, L., 2016. Interaction between warm Atlantic-sourced waters and the East Greenland Current in northern Denmark Strait (68°N) during the last 10 600 cal a BP. *Journal of Quaternary Science* 31, 472-483.

Perner, K., Moros, M., Lloyd, J.M., Jansen, E., Stein, R., 2015. Mid to late Holocene strengthening of the East Greenland Current linked to warm subsurface Atlantic water. *Quaternary Science Reviews* 129, 296-307.

Perner, K., Moros, M., Lloyd, J.M., Kuijpers, A., Telford, R.J., Harff, J., 2011. Centennial scale benthic foraminiferal record of late Holocene oceanographic variability in Disko Bugt, West Greenland. *Quaternary Science Reviews* 30, 2815-2826.

- Seidenkrantz, M.-S., Roncaglia, L., Fischel, A., Heilmann-Clausen, C., Kuijpers, A., Moros, M., 2008. Variable North Atlantic climate seesaw patterns documented by a late Holocene marine record from Disko Bugt, West Greenland. *Marine Micropaleontology* 68, 66-83.
- Seidenkrantz, M.S., Aagaard-Sørensen, S., Sulsbrück, H., Kuijpers, A., Jensen, K.G., Kunzendorf, H., 2007. Hydrography and climate of the last 4400 years in a SW Greenland fjord: implications for Labrador Sea palaeoceanography. *The Holocene* 17, 387-401.
- Sha, L., Jiang, H., Seidenkrantz, M.-S., Knudsen, K.L., Olsen, J., Kuijpers, A., Liu, Y., 2014. A diatom-based sea-ice reconstruction for the Vaigat Strait (Disko Bugt, West Greenland) over the last 5000yr. *Palaeogeography, Palaeoclimatology, Palaeoecology* 403, 66-79.
- Simstich, J., Sarnthein, M., Erlenkeuser, H., 2003. Paired $\delta^{18}\text{O}$ signals of *Neogloboquadrina pachyderma* (s) and *Turborotalita quinqueloba* show thermal stratification structure in Nordic Seas. *Marine Micropaleontology* 48, 107-125.
- Straneo, F., Kawase, M., Pickart, R.S., 2002. Effects of wind on convection in strongly and weakly baroclinic flows with application to the Labrador Sea. *Journal of Physical Oceanography* 32, 2603-2618.
- Thornalley, D.J.R., Elderfield, H., McCave, I.N., 2009. Holocene oscillations in temperature and salinity of the surface subpolar North Atlantic. *Nature* 457, 711-714.
- Werner, K., Frank, M., Teschner, C., Müller, J., Spielhagen, R.F., 2014. Neoglacial change in deep water exchange and increase of sea-ice transport through eastern Fram Strait: evidence from radiogenic isotopes. *Quaternary Science Reviews* 92, 190-207.
- Werner, K., Müller, J., Husum, K., Spielhagen, R.F., Kandiano, E.S., Polyak, L., 2016. Holocene sea subsurface and surface water masses in the Fram Strait – Comparisons of temperature and sea-ice reconstructions. *Quaternary Science Reviews* 147, 194-209.
- Yashayaev, I., 2007. Hydrographic changes in the Labrador Sea, 1960-2005. *Progress In Oceanography* 73, 242-276.

REVIEWERS' COMMENTS:

Reviewer #2 (Remarks to the Author):

The authors have revised the manuscript appropriately and have addressed the main concerns.

Some figures and captions have been improved and the addition of the 5th figure (suggested by Review#1) clarifies the general understanding of the warm/cold climate over Europe, as this paper is intended to a broad readership. I only suggest to move "cold" a little away from the tip of the arrow (weak heat flow) as it may be misunderstood as cold flowing from the tropics.

In figure 4, the authors have well synthesized and linked the major historical events that occurred in Europe with the disruptions of tropical heat transport to the North as presented in the data. The figure does not force conclusions but rather gives good indications on the history linked to climate changes. Although this may not present a novelty, it adds one step towards the comprehension of the climatic shifts that may have triggered or exacerbated some historical events.

A few editing suggestions:

line 97: as this paper is intended for broad readership, maybe a small explanation of what foraminifera are (unicellular marine micro-organism),

line 98: (s) sinistral coiling

Reviewer #3 (Remarks to the Author):

Moffa-Sanchez and Hall have successfully addressed my two most important critiques, surrounding the workings of the Dd18O proxy not originally explained, and the highlighting of the sortable silt record that initially fell through the cracks. While the importance of wind and turbulent buoyancy forcing might still be underestimated, the discussion of that possibility is improved. I recommend reiterating around line 146 that the import of cold and fresh polar waters acts to precondition weakened winter convection, such that salinity has a bigger impact than temperature: i.e. the cold water does not aid convection because the winter always gets cold enough in this framework; this would clarify a potentially counterintuitive association (cold = weak convection) for some readers. Also remove the word "convincing" from line 252 as that is up to the opinion of the reader. I find the revised manuscript to be of sufficient quality and importance to publish in Nature Communications.